# Risk of Dementia According to Surgery Type: A Nationwide Cohort Study

**DOI:** 10.3390/jpm12030468

**Published:** 2022-03-15

**Authors:** Young Suk Kwon, Sang-Hwa Lee, Chulho Kim, Hyunjae Yu, Jong-Hee Sohn, Jae Jun Lee, Dong-Kyu Kim

**Affiliations:** 1Division of Big Data and Artificial Intelligence, Institute of New Frontier Research, Chuncheon Sacred Heart Hospital, Hallym University College of Medicine, Chuncheon 24253, Korea; gettys@hallym.or.kr (Y.S.K.); neurolsh@hallym.or.kr (S.-H.L.); gumdol52@hallym.or.kr (C.K.); yunow@hallym.or.kr (H.Y.); iloveu59@hallym.or.kr (J.J.L.); 2Department of Anesthesiology and Pain Medicine, Chuncheon Sacred Heart Hospital, Hallym University College of Medicine, Chuncheon 24253, Korea; 3Department of Neurology, Chuncheon Sacred Heart Hospital, Hallym University College of Medicine, Chuncheon 24253, Korea; 4Department of Otorhinolaryngology-Head and Neck Surgery, Chuncheon Sacred Heart Hospital, Hallym University College of Medicine, Chuncheon 24253, Korea

**Keywords:** surgery type, dementia, Alzheimer’s disease, vascular dementia, anesthesia, incidence rate

## Abstract

The relationship between dementia and surgery remains unclear. Research to elucidate the relationship between them is scarce, and conducting epidemiological research is complicated. This study aimed to investigate the incidence and risk of dementia according to the surgery type. We performed a retrospective propensity score-matched cohort study using nationwide representative cohort sample data from the Korean National Health Insurance Service in South Korea between 2003 and 2004. Incidence rates for dementia were obtained by dividing the number of patients with dementia by person-years at risk. To identify the risk of dementia according to the type of surgery, we investigated the hazard ratio by each surgery type. The incidence rates of dementia in control, musculoskeletal, and two or more surgeries groups were 9.66, 13.47, and 13.36 cases per 1000 person-years, respectively. The risk of dementia in the musculoskeletal and two or more surgeries groups was 1.44-fold higher (95% confidence interval (95% CI), 1.22–1.70) and 1.42-fold higher (95% CI, 1.17–1.72) than that in the control group, respectively. Patients who underwent musculoskeletal surgery and two or more surgeries had a higher risk of dementia; however, there was no association with the type of anesthesia administered.

## 1. Introduction

Dementia is currently ranked the seventh cause of death among all diseases and is one of the main causes of disability and dependence among the elderly worldwide. At present, more than 55 million dementia patients are reported worldwide. Every year, dementia develops in about 10 million patients. Dementia is a deterioration in cognitive function beyond the usual consequences of biological aging [1]. A variety of diseases and injuries affecting the brain may cause dementia.

Surgery and anesthesia may affect the brain, as well as immediate survival and long-term outcomes [2,3]. Many studies have reported an association between anesthesia and dementia; however, this association remains controversial [4,5,6,7,8,9,10,11,12]. Anesthesia usually involves a series of surgical processes; therefore, it is difficult to evaluate anesthesia and surgery for dementia separately. Surgery is a very stressful process for dementia patients [13] and postoperative care is usually much longer. Thus, surgery affects the postoperative status of patients more than anesthesia and may be more important in evaluating the risk of dementia than anesthesia. In particular, because reducing activities increases the risk of dementia [14] and postoperative care such as ambulation, immobility, and rehabilitation are significantly different according to the surgical type, surgical types may be important in evaluating the risk of dementia. Therefore, we hypothesized that the surgical type could affect the occurrence of dementia. We evaluated the incidence and risk of dementia according to the surgical type in this study and included Alzheimer’s disease and vascular dementia, which are common subtypes of dementia.

## 2. Materials and Methods

### 2.1. National Sample Cohort

South Korea has provided compulsory health coverage to people from 1989 through the Korean National Health Insurance Service (KNHIS). A registration number assigned from birth protects healthcare system data from omission and duplication. In 2006, medical aid data and KNHIS were integrated, and the combined data are free from selection bias because it captures all South Korean residents. It contains extensive data, including qualifications and insurance premiums, health checkup results, medical records, long-term care insurance data for the elderly, nursing home status, and registration information for cancer and rare, incurable diseases. This study used a representative sample from the 2002 to 2013 National Sample Cohort in South Korea (NHIS-2018-2-258). The dataset included approximately 2% (1,025,340 adults) of the South Korean population in 2002. Stratified random sampling was performed using 1476 strata by age (5-year intervals, 18 groups), sex (male and female, two groups), and income level (40 health insurance groups and one medical aid beneficiary, 41 groups) of the South Korean population (46 million) in 2002.

### 2.2. Study Setting and Participants

This study was conducted after obtaining approval from our institutional review board. Since the KNHIS-NSC dataset is created for research purposes, it consists of unidentified secondary data. Hence, the need for written informed consent was exempted. We set the index period from 1 January 2003 to 31 December 2004 and included a study cohort of elderly people aged 55 years or older who underwent surgery under general anesthesia or neuraxial anesthesia during the index period. We set a washout period from 1 January 2002 to 31 December 2002 to eliminate cases that involved surgeries under anesthesia before the index period. We excluded patients who (1) underwent surgery under general or neuraxial anesthesia before and after the index period, (2) underwent surgery with anesthesia other than general and neuraxial anesthesia from 2002 to 2013, (3) underwent heart or brain surgery between 2002 and 2013, (4) were diagnosed with dementia before and during the index period, or (5) died during the index period. The comparison group, which consisted of patients who did not undergo surgery under general or neuraxial anesthesia, was randomly selected from the remaining cohorts enrolled in the database between 2003 and 2004 (one for each patient who underwent surgery under general or neuraxial) with propensity score-matched individuals. Both groups of this study were matched by sex, age, residence, residential household income, Charlson comorbidity index (CCI), and the year of enrolment. A schematic description of the cohort composition is summarized in Figure 1.

### 2.3. Predictor and Outcome Variables

We collected data from patients diagnosed with dementia. The diagnosis was determined by a code using the Korean Standard Classification of Disease (KCD). Since KCD is derived from the International Classification of Diseases (ICD), it is similar to ICD [15,16]. Alzheimer’s disease and vascular dementia were also diagnosed using the same method as that of dementia. The codes used were as follows: Alzheimer’s disease, F00, G30; vascular dementia, F01; and others, F02, F03 (Appendix A Table A1). The study participants were divided into three categories according to age (55–64, 65–74, and ≥75 years), residence (1st area: Seoul, the largest metropolitan region in South Korea; 2nd area: other metropolitan cities in South Korea; and 3rd area: small cities and rural areas), household income (low: ≤30%, middle: 30.1–69.9%, and high: ≥70% of the median), and CCI (0, 1, and ≥2). CCI is a weighted index to predict the risk of death within 1 year of hospitalization for patients with specific comorbid conditions [17]. It was developed based on medical records and converted into ICD-10 codes for 19 diseases to be applied as administrative data. Appendix A Table A2 presents a list of Charlson comorbidities, ICD-10 code, and original and updated weights. The endpoint of the study was patient death or the development of dementia. Appendix A Table A3 shows the description of time to the event and censored data of all patients. The types of surgery were classified into nine categories: musculoskeletal system, skin and soft tissue surgery, digestive surgery, genitourinary surgery, obstetrics and gynecologic surgery, neurosurgery, circulatory surgery, and others. The others category included patients who underwent surgeries except for the aforementioned seven surgery types under general or neuraxial anesthesia. Moreover, patients who underwent two or more surgeries were assigned to the two or more surgeries category.

### 2.4. Statistical Analysis

We applied one-to-one propensity score matching according to age, sex, residence area, household income, and CCI. The incidence rates per 1000 person-years for dementia were calculated by dividing the number of patients with dementia by person-years at risk. To identify whether surgical types increased the risk of occurrence of dementia, we used Cox proportional hazards regression analyses to calculate the hazard ratio (HR) and 95% confidence intervals (CI), adjusting for other outcome variables. The overall dementia-free survival rate was determined using Kaplan–Meier survival curves for the follow-up period (the *p*-value was used to reject the null hypothesis, i.e., HR = 1). To evaluate the risk association between each surgical type and dementia, we used Cox proportional hazard regression to calculate the HR and 95% CI while adjusting for other co-variables. Similarly, the incidence and risk of Alzheimer’s disease and vascular dementia as subtypes of dementia were also investigated. For subgroup analysis, we evaluated the HRs of dementia according to sex and the type of dementia among the matched patients. In addition, the incidence and risk of dementia were calculated according to the type of anesthesia and scale of surgery in surgical types with a statistically significant risk of developing dementia. All statistical analyses were performed with R (version 3.4.3; R Foundation for Statistical Computing, Vienna). Two-sided *p* < 0.05 was considered statistically significant.

## 3. Results

This study consisted of 7296 patients who underwent surgery under neuraxial or general anesthesia and 7296 comparison participants who did not undergo surgery under neuraxial or general anesthesia. The two cohorts had equal distributions of age, residence, household income, and CCI and similar distribution of sex, indicating consistency between the two groups (Table 1, Appendix A Figure A1).

### 3.1. Hazard Ratios of Dementia According to Surgery Type

Figure 2 represents the Kaplan–Meier survival curves with log-rank tests for the cumulative hazard plot of specific dementia-free status between the comparison of each surgery type. The results of the log-rank test indicated that patients who underwent musculoskeletal surgery and two or more surgeries developed dementia more frequently than those who did not undergo surgery under anesthesia during the 9-year follow-up period. In the subgroup analysis, patients who underwent musculoskeletal surgery and two or more surgeries developed Alzheimer’s disease more frequently than those who did not undergo surgery under anesthesia during the 9-year follow-up period, but there were no differences in the incidence of vascular dementia (Figure 3).

The overall incidence of dementia was 9.66 per 1000 person-years in the control group, compared to more than 13 per 1000 person-years in the musculoskeletal, skin and soft tissue, circulatory, and two or more surgeries groups. Simple and multiple Cox regression models were used to analyze the HR for the development of dementia (Table 2). After adjusting for socio-demographic factors and Charlson comorbidities, we found that musculoskeletal surgery (HR, 1.44; 95% CI, 1.22–1.70) and two or more surgeries (HR, 1.42; 95% CI, 1.17–1.72) were associated with prospective dementia development.

### 3.2. Subgroup Analysis According to Sex, Dementia Type, Anesthesia Type, and Scale of Surgery

In the subgroup analysis according to sex, musculoskeletal surgery (male, adjusted HR (95% CI), 1.56 (1.17–2.09); female sex, adjusted HR (95% CI), 1.38 (1.13–1.69)) demonstrated a significant risk of developing dementia for both males and females. In two or more surgeries, only females had a significantly higher risk of developing dementia (adjusted HR (95% CI), 1.47 (1.16–1.87)) (Appendix A Table A4). The incidence of Alzheimer’s disease with musculoskeletal surgery and two or more surgeries were 10.46 and 10.27 per 1000 person-year (reference: control, 7.01), respectively. The risk of developing Alzheimer’s disease with musculoskeletal (adjusted HR (95%CI), 1.55 (1.28–1.86)) and two or more surgeries (adjusted HR (95%CI), 1.50 (1.21–1.88)) was significantly higher than those in the control group (Appendix A Table A5). There were no significant differences in the risk of developing vascular dementia according to surgical type.

When the risk of developing dementia after musculoskeletal surgery and two or more surgeries was compared between general anesthesia and neuraxial anesthesia, the risk of developing dementia, Alzheimer’s disease, and vascular dementia was absent, regardless of matching (*p* >0.05). The incidence and HRs according to anesthesia type are summarized in Appendix A Table A6 and Table A7.

In musculoskeletal surgery, when the risk of developing dementia was compared with the scale of surgery (minor and major surgery type) to that of the control group, there was an increase in the risk of developing dementia (minor, adjusted HR (95% CI), 1.51 (1.19–1.93); major, adjusted HR (95% CI), 1.56 (1.16–2.11)) and Alzheimer’s disease (minor, adjusted HR (95% CI), 1.60 (1.21–2.11); major, adjusted HR (95% CI), 1.68 (1.20–2.35)). The incidence and HRs according to the surgery scale are summarized in Appendix A Table A8 and Table A9.

## 4. Discussion

The primary aims of this study were to evaluate if the incidence and risk of dementia were related to surgery type. Of the cohort that increased risk of dementia after surgery, the secondary aims of this study were to (1) evaluate if the incidence and risk of Alzheimer’s disease or vascular dementia—the two most common dementia subtypes—were higher relative to a generalized population and (2) determine the interplay of anesthesia type and surgery scale on dementia. The incidence of dementia in the control group during the 9-year follow-up period was 9.66 cases per 1000 person-years, and among those with musculoskeletal and two or more surgeries were 13.47 and 13.36 cases per 1000 person-years, respectively. The incidence of risk of dementia in older adults who underwent musculoskeletal surgery and two or more surgeries was 1.44-fold and 1.42-fold higher than that in older adults who did not undergo surgery under anesthesia, respectively. We observed this even after adjusting for several risk factors. Similarly, the incidence risk of Alzheimer’s disease in older adults who underwent musculoskeletal surgery and two or more surgeries was higher than that in older adults who did not undergo surgery under anesthesia.

The relationship between anesthesia, surgery, and dementia remains controversial [4,6,7,9,12,18,19,20,21,22,23,24,25,26,27]. Several nationwide studies have reported that anesthesia or surgery increases the risk of dementia [6,9,20,24,25]; however, the results of meta-analyses were inconsistent [7,22,23]. It is very difficult to investigate the relationship between dementia and surgery type or anesthesia alone. In the clinical environment and human research, anesthesia and surgery remain inseparable. It is also difficult and unethical to perform prospective, large-scale, sample-size randomized controlled trials to evaluate the association between anesthesia and/or surgery and dementia [23,27]. Therefore, previous studies are mostly observational, retrospective studies [6,23]. Moreover, inconsistent results about the risk of dementia, in addition to the lack of sufficient large-scale randomized controlled trials, would inevitably affect several meta-analyses and thereby impact the interpretation of the results of the present study.

Animal and laboratory experiments have reported that some anesthetics cause pathological changes in beta-amyloid deposition and tau hyperphosphorylation associated with dementia [28,29,30,31,32,33,34]. However, clinically, it remains unknown whether anesthesia itself induces or accelerates structural changes in the brain, which causes cognitive decline. Modern anesthetics have short acting periods and a context-sensitive half-time [35]. Several studies have found a link between drugs used for anesthesia and cognitive impairment in the elderly, but the effect is presumed to be temporary and reversible [36]. Additionally, another study showed that the direct effect of regional anesthesia on the brain might be limited when comparing general anesthesia and regional anesthesia [26].

All surgery types have a specificity, but it is unknown what are the effects of specificity of the surgical procedure on dementia, except for studies related to heart surgery [37,38]. In our results, we could not determine whether the surgical procedure itself, events after the surgery, or both increased the risk of dementia. However, the specificity of musculoskeletal surgery differs from that of other surgeries in postoperative care. Most patients undergoing musculoskeletal surgery use casts or devices to prevent mobility for a long time [39]. Hip fractures occur primarily in the elderly. Most patients with hip fractures experience reduced mobility and lose their ability to function independently. Approximately 45% of community residents at the time of a fracture are discharged to the institution after being hospitalized, and 15–25 % remain in the institution for 1 year after the fracture. In addition, the activities of patients may be limited during rehabilitation after wound recovery [40]. Further, the recovery and rehabilitation periods can be greatly delayed in old age [41]. Such limited activities are an important risk factor for dementia [14]. Using an orthopedic surgery mouse model, a report provided translational evidence for a role for peripheral surgery in contributing to delirium-like behavior and disrupted neuro-immunity. In the same study, mice showed time-dependent attention-deficit impairment after surgery and a return to preoperative performance on day 5, postoperatively. However, because this was not an animal model for dementia, and the mice model was not old and lacked long-term follow-up, it was difficult to determine the association between musculoskeletal surgery and dementia [42]. It is also difficult to determine the cause of the increased risk of Alzheimer’s disease in two or more surgeries. Repetitive exposure to anesthetics and surgical stress, reducing activities due to repetitive surgery, and postoperative complications can be potential causes. However, in this study, because the category of two or more surgeries included heterotypic surgeries, it was difficult to determine the repetitive effect of detailed type of surgery on dementia.

The present study investigated the risk of dementia, Alzheimer’s disease, and vascular dementia according to the surgery type since the pathophysiology of each condition is different, and we observed a difference in the results. Alzheimer’s disease is the most common type of dementia, followed by vascular dementia [43], and is a neurodegenerative and remarkable protein-conformational disease [44,45] caused primarily by abnormal processing and polymerization of normally soluble proteins [46]. Vascular dementia refers to any form of dementia that is predominantly caused by cerebrovascular disease or cerebral blood flow disorder [47]. Such differences may be associated with our results. Stroke can cause vascular dementia; therefore, the risk factor for stroke is also a risk factor for vascular dementia in the general population. The main risk factors for vascular dementia in population-based studies are advanced age, hypertension, diabetes, elevated total cholesterol levels, lower physical activity, low or high body mass index, smoking, coronary artery disease, and atrial fibrillation [48,49,50,51,52,53,54,55,56]. After matching patient characteristics, because the possibility of differences in main risk factors according to surgical type is reduced, there might not be a difference in the risk of vascular dementia according to surgical type. Postoperative cerebrovascular accidents occurred late in the postoperative period, i.e., 5–26 days after surgery, and it has been reported that postoperative cerebrovascular accidents are not directly related to surgery and anesthesia [57].

The present study has some limitations. First, we could not determine the anesthetic and surgical history of patients before the washout period because they were divided according to the presence of surgery under anesthesia experience during the index period. Considering the slow progression of dementia, anesthesia and surgery experience before the washout period may have affected the development of dementia. Therefore, to reduce information bias as much as possible, we only enrolled participants who were 55 years of age or older and investigated the effect of surgery under general or neuraxial anesthesia in old age. Additionally, this study had a 1-year washout period for surgery under anesthesia and excluded patients with additional surgery under anesthesia after the index period to ensure that only the effect of surgery under general or neuraxial anesthesia during the index period was evaluated. Moreover, we matched the surgery under general or neuraxial anesthesia and the comparison using propensity scores. Second, although we classified the surgery type and anesthesia type in this study, it was difficult to evaluate the detailed effect of various surgeries and local anesthesia on our findings. The scale of surgery and perioperative events, including complications, may affect immediate survival and long-term outcomes [33]. However, the KNHIS-NSC dataset has been established for medical service claims and reimbursement but not for research. This dataset lacked detailed data on surgeries and local anesthesia. Third, our results could be affected by various biases, including the frailty of patients, preference of patients or physician for anesthesia, and various events that affect the brain during the follow-up period. Moreover, there may be further bias in patients who underwent two or more surgeries.

In conclusion, musculoskeletal surgery and two or more surgeries increased the risk of dementia and Alzheimer’s disease. However, in the subgroup analysis, musculoskeletal surgery and two or more surgeries increased the risk of Alzheimer’s disease but not the risk of vascular dementia. Although we cannot determine the detailed cause, considering that there was no difference depending on the type of anesthesia, we can consider anesthesia, the surgery itself, or serial effects due to surgery as the risk factors for dementia and Alzheimer’s disease. Our results present new considerations for patients and clinicians regarding the risk of dementia according to surgery type and a new perspective for selecting the preferred anesthesia type in high-risk patients with dementia. However, as the results were different for different subtypes of dementia, further studies are needed.

## Figures and Tables

**Figure 1 jpm-12-00468-f001:**
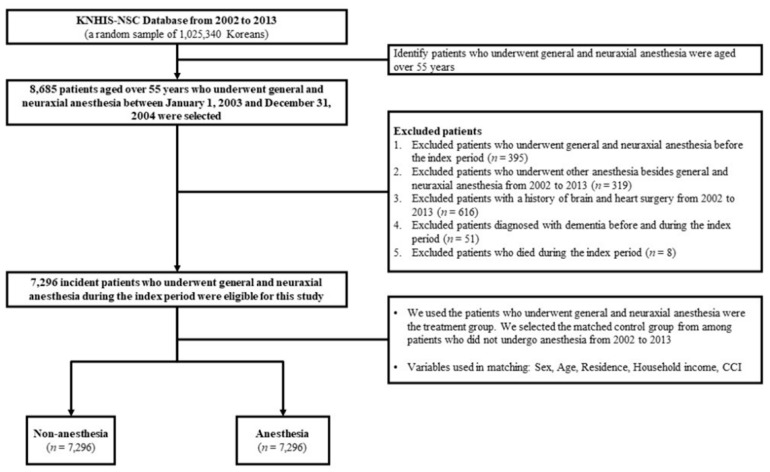
Flow chart of study design. CCI—Charlson comorbidity index.

**Figure 2 jpm-12-00468-f002:**
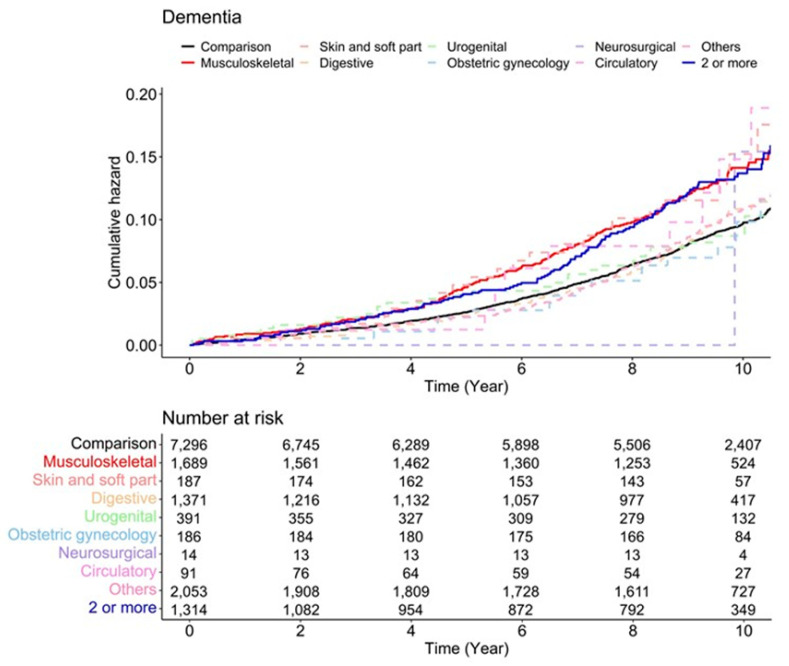
Risk of development of dementia according to surgery type.

**Figure 3 jpm-12-00468-f003:**
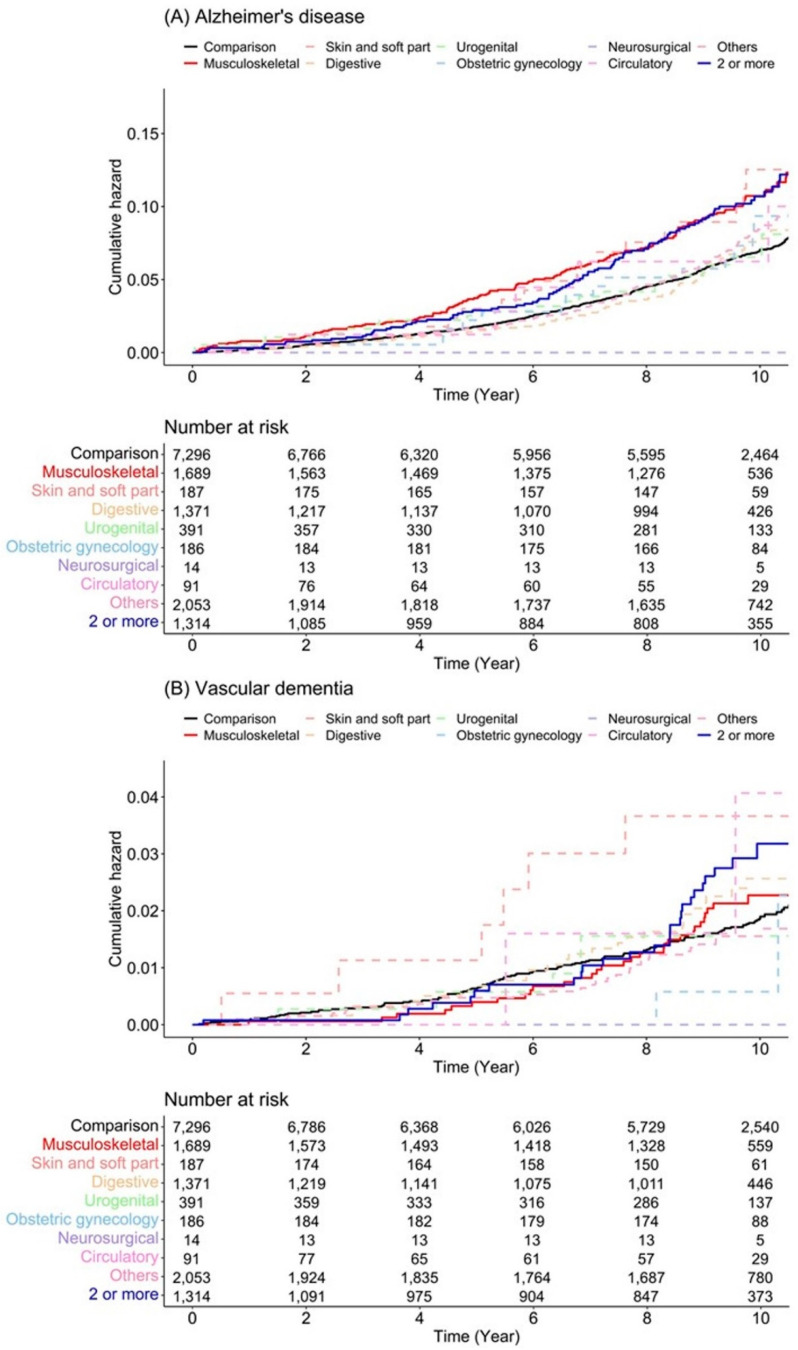
Cumulative hazard plot of specific disease according to surgery type: (**A**) Alzheimer’s disease and (**B**) vascular dementia.

**Table 1 jpm-12-00468-t001:** Characteristics of the cohort.

Variables	Comparison (*n* = 7296)	Surgery under Anesthesia (*n* = 7296)	*p*-Value
**Sex**			1.000
Male	3303 (45.3%)	3304 (45.3%)	
Female	3993 (54.7%)	3992 (54.7%)	
**Age (years)**			1.000
55–64	3636 (49.8%)	3635 (49.8%)	
65–74	2667 (36.6%)	2668 (36.6%)	
≥75	993 (13.6%)	993 (13.6%)	
**Residence**			1.000
Seoul	1574 (21.6%)	1574 (21.6%)	
Second area	1522 (20.9%)	1522 (20.9%)	
Third area	4200 (57.6%)	4200 (57.6%)	
**Household income**			1.000
Low (0–30%)	1489 (20.4%)	1489 (20.4%)	
Middle (30–70%)	2377 (32.6%)	2377 (32.6%)	
High (70–100%)	3430 (47.0%)	3430 (47.0%)	
**CCI**			1.000
0	3273 (44.9%)	3273 (44.9%)	
1	1358 (18.6%)	1358 (18.6%)	
≥2	2665 (36.5%)	2665 (36.5%)	

Comparison—subjects without anesthesia; Seoul—the largest metropolitan area; second area—other metropolitan cities; third area—other areas; CCI—Charlson comorbidity index.

**Table 2 jpm-12-00468-t002:** Incidence per 1000 person-years and HR (95% CIs) of dementia during the follow-up period.

Surgical Site	N	Case	Incidence	Unadjusted HR (95% CI)	Adjusted HR (95% CI)	*p*-Value
Comparison	7296	615	9.66	1.00 (ref)	1.00 (ref)	
Musculoskeletal	1689	189	13.47	1.46 (1.24–1.72) *	1.44 (1.22–1.70) *	<0.001
Skin and soft part	187	22	13.99	1.51 (0.99–2.31)	1.51 (0.99–2.32)	0.057
Digestive	1371	107	9.75	1.06 (0.86–1.30)	1.03 (0.84–1.27)	0.776
Urogenital	391	29	9.13	0.99 (0.68–1.44)	0.89 (0.61–1.30)	0.561
Obstetric gynecology	186	16	9.08	0.95 (0.58–1.56)	0.89 (0.54–1.47)	0.653
Neurosurgical	14	1	7.80	0.82 (0.11–5.81)	1.42 (0.20–10.10)	0.727
Circulatory	91	9	13.87	1.55 (0.80–2.99)	1.92 (0.99–3.72)	0.052
Others	2053	182	10.32	1.10 (0.94–1.30)	1.14 (0.97–1.35)	0.114
2 or more	1314	125	13.36	1.48 (1.22–1.80) *	1.42 (1.17–1.72) *	<0.001

HR—hazard ratio; CI—confidence interval. * *p* < 0.001.

## Data Availability

Data can be accessed through the NHIS’ National Health Insurance Data Sharing Service website [http://nhiss.nhis.or.kr/bd/ab/bdaba021eng.do, accessed on 9 March 2022].

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
