# Peer review of "Risk of Dementia According to Surgery Type: A Nationwide Cohort Study"

_jpm, 2022, doi:10.3390/jpm12030468_

Round 1

Reviewer 1 Report

The retrospective study by young suk kwon et al. identified the risk of developing dementia according to the surgery type in the Korean population dataset. This manuscript is very similar and several phrases are common to the previous study (https://doi.org/10.3390/jpm11121386). Nevertheless, It suggests musculoskeletal surgery and two or more surgeries could be risk factors for dementia.  However, it needs further developments/clarifications before acceptance.

Major comments,

  • Will the risk of musculoskeletal surgery and two or more surgeries for dementia will change/remain the same if I choose any varied duration of one year period and select different patients from KNHIS -NSC. For example surgeries from years 2004 to 2005 ( and do 8 -year follow-up) or surgeries from 2005 to 2006 ( and do 7-year follow-up).
  • Is there any experimental evidence in the animal model for musculoskeletal surgery and dementia? If available kindly do discuss in the discussion.
  • In the methods section (or in the table description ) describe the type of statistical test performed to obtain a p-value.

Minor comments,

Check for grammar, mismatch, missing words, phrases.

  • In lines 18-19, This study aimed to “investigated” incidence of dementia
  • In line 24 “type. the incidence rates”
  • Table 1 description – add third area
  • Figure 3 description – Cumulative hazard plot of specific according to surgery type
  • Figure 3 description – Change “vascular disease” to “Vascular dementia
  • In line 217 – “in the use of drugs used for anesthesia”

Author Response

Reviewer 1.

  1. Comment

Will the risk of musculoskeletal surgery and two or more surgeries for dementia will change/remain the same if I choose any varied duration of one year period and select different patients from KNHIS -NSC. For example surgeries from years 2004 to 2005 ( and do 8 -year follow-up) or surgeries from 2005 to 2006 ( and do 7-year follow-up).

Answer

Thank you for your comments. Unfortunately, we could not further cohort sampling at this time, because this public dataset was needed permission to utilize every time. However, in this study, we presented the results of the log-rank test as the number of events according to the follow-up period (2, 4, 6, 8, and 10 years) in Figures 2 and 3. So, we thought these findings could be explainable to the reviewer's comments. Again, thank you for your understanding.

  1. Comment

Is there any experimental evidence in the animal model for musculoskeletal surgery and dementia? If available kindly do discuss in the discussion.

Answer:

Thank you for your comments.  However, to date, there has been no experimental evidence in animal models for musculoskeletal surgery and dementia. Only, there was an orthopedic surgery mouse model associated with attention deficit impairment, so we added it.

Using an orthopedic surgery mouse model, a report provided translational evidence for a role for peripheral surgery in contributing to delirium-like behavior and disrupted neuroimmunity. In the same study, mice showed time-dependent attention deficit impairment after surgery and a return to preoperative performance on day 5, postoperatively. However, because this was not an animal model for dementia, and the mice model were not old and lacked long-term follow-up, it was difficult to determine the association between musculoskeletal surgery and dementia.

  1. Comment

In the methods section (or in the table description ) describe the type of statistical test performed to obtain a p-value.

Answer:

As you commented, we added the description of the type of statistical test in the section of Method.

  1. Comment

In lines 18-19, This study aimed to “investigated” incidence of dementia

Answer:

Thank you for your comment. We correct it.

  1. Comment

In line 24 “type. the incidence rates”

Answer:

Thank you for your comment. We correct it.

  1. Comment

Table 1 description – add third area

Answer:

Thank you for your comment. We added description of third area.

  1. Comment

Figure 3 description – Cumulative hazard plot of specific according to surgery type

Answer:

Thank you for your comment. We added description of Figure 3

  1. Comment

Figure 3 description – Change “vascular disease” to “Vascular dementia”

Answer:

Thank you for your comment. We correct it.

  1. Comment

In line 217 – “in the use of drugs used for anesthesia”

Answer:

Thank you for your comment. We correct it.

Reviewer 2 Report

Main concern

I am confused by the authors use of the terms dementia, Alzheimer’s disease, and vascular dementia throughout the manuscript. The authors refer to ‘dementia’ (only) in the Introduction. Then, in the Methods, Results, and Discussion, the authors demarcate the term ‘dementia’ from that of Alzheimer’s disease and vascular dementia.

Dementia is an umbrella term that refers to many dementia subtypes. Alzheimer’s disease and vascular dementia are forms of dementia. Other types of dementias include frontotemporal dementia, Lewy body dementia and Parkinson’s plus syndromes (i.e., progressive supranuclear palsy, corticobasal syndrome, multiple systems atrophy). At this point, I am unsure what this manuscript is analysing. What types of dementias are in the ‘dementia’ group (i.e., Alzheimer’s disease, vascular dementia, frontotemporal dementia, Lewy body dementia)? Are the Alzheimer’s disease and vascular dementia patients in this study also included in the ‘dementia’ group? If this study is, in fact, a comparison between Alzheimer’s disease and vascular dementia, should this be stated in the Introduction aims and concluding remarks of the Discussion?

Other points

Abstract, line 19: “This study aimed to investigated incidence…”. Change to either: i) “This study investigated the incidence of dementia”, or ii) This study aimed to investigate the incidence of dementia”.

Abstract, line 24: Capitalise the beginning of the sentence. “… of each surgery type. The incidence rates of dementia…

Line 35: Provide a reference for the claim that dementia is the 7th most common cause of death.

Line 37: Provide a reference for the claim that there are 55 million dementia patients worldwide etc.

Line 38: The full stop should occur before the [1] reference.

Figure 1. and Table 1. CCI has not been defined. Either the Charleson comorbidity index should be spelled out in the figure/table, or an abbreviations caption should be provided below the figure/table.

Line 95: The International Classification of Diseases (ICD) appears to be an established assessment tool. Further, it appears the authors are using the 10th revision of the ICD? Please reference accordingly.

Line 98: I am unfamiliar with the terms F00, G30, F02, F03. What does the F stand for? What do the values 00, 30, 02, 03 stand for? Could the authors please provide more information about these coding values?

Line 102-103: The Charlson comorbidity index appears to be an established assessment tool. Please reference accordingly.

Line 105 and Appendix Table A1: Can the authors provide more information on the ‘original’ and ‘updated weights’ in Appendix Table A1? Is the original value referring to the ICD-10 and the updated weight referring to the KCD? Further can the authors justify why the weights were updated? For example, what was the reasoning for revising the peripheral vascular disease classifications from a 1 (original weight) to 0 (updated weight)?

Line 107 and Appendix Table A2: The authors claim Appendix Table A2 shows the description of ‘time to event’. I am unsure which values refer to ‘time’? Should a time value be provided?

Appendix Table A2. It appears this table refers to the i) total number of surgical events, ii) the number of patients who developed a dementia following surgery, categorised according to surgical event, iii) the number of patients who subsequently died, categorised according to surgical event, and iv) the number of patients who were not able to be followed up, categorised according to surgical event? If I have understood this table correctly, then this table could be improved if the categories went across the page (see example attached) and the table provided the number and proportional values (i.e., as a percentage of the total value: e.g., 63% etc). Can the authors clarify the heading of the second column (i.e., “The number of dementia event”) of Table A2? This would imply that all numbers in the second column refer to patients developing dementia?

The authors could consider redesigning the table the following way? (see attached)

Lines 108-110. The authors state: “The types of surgery were classified into nine categories: musculoskeletal system, skin and soft tissue, digestive surgery, genitourinary surgery, obstetrics and gynecologic surgery, neurosurgery, and circulatory surgery.” There are two missing categories in this sentence: ‘non-anesthesia’ and ‘others’? The authors should define ‘others’ here. The authors should consider bullet pointing the categories. For example, “The types of surgery were classified into nine categories: i) non-anesthesia, ii) musculoskeletal system, iii) etc”.

Line 203. There is a double space between ‘were inconsistent’.

Line 315. Appendix Table A3. Consider starting the table title on a new page. It is disconnected from the table.

Line 337. The References title should be on a new page.

Author Response

Reviewer 2

  1. Comment

Main concern

I am confused by the authors use of the terms dementia, Alzheimer’s disease, and vascular dementia throughout the manuscript. The authors refer to ‘dementia’ (only) in the Introduction. Then, in the Methods, Results, and Discussion, the authors demarcate the term ‘dementia’ from that of Alzheimer’s disease and vascular dementia.

Dementia is an umbrella term that refers to many dementia subtypes. Alzheimer’s disease and vascular dementia are forms of dementia. Other types of dementias include frontotemporal dementia, Lewy body dementia and Parkinson’s plus syndromes (i.e., progressive supranuclear palsy, corticobasal syndrome, multiple systems atrophy). At this point, I am unsure what this manuscript is analysing. What types of dementias are in the ‘dementia’ group (i.e., Alzheimer’s disease, vascular dementia, frontotemporal dementia, Lewy body dementia)? Are the Alzheimer’s disease and vascular dementia patients in this study also included in the ‘dementia’ group? If this study is, in fact, a comparison between Alzheimer’s disease and vascular dementia, should this be stated in the Introduction aims and concluding remarks of the Discussion?

Answer:

Thank you for good comment. The aim of this study was to investigate incidence and risk of dementia in patients who underwent surgery under anesthesia. Dementia has many subtypes like you said. So, we analyzed common subtypes (Alzheimer's disease and vascular dementia) of dementia, additionally. Also, in KCD code, dementia except for Alzheimer's disease and vascular disease was classified to dementia in other diseases classified elsewhere or unspecified dementia.

We added description in the introduction aim and discussion.

Introduction

We evaluated the incidence and risk of dementia according to the surgical type in this study, and included Alzheimer's disease and vascular dementia, which are common subtypes of dementia..

Method— Statistical analysis

Similarly, the incidence and risk of Alzheimer's disease and vascular dementia as subtypes of dementia were also investigated.

Discussion

In conclusion, musculoskeletal surgery and two or more surgeries increased the risk of dementia and Alzheimer's disease. However, in the subgroup analysis, musculoskeletal surgery and two or more surgeries increased the risk of Alzheimer's disease but not the risk of vascular dementia. Although we cannot determine the detailed cause, considering that there was no difference depending on the type of anesthesia, we can consider anesthesia, the surgery itself, or serial effects due to surgery as the risk factors for dementia and Alzheimer's disease. Our results present new considerations for patients and clinicians regarding the risk of dementia according to surgery type, and a new perspective for selecting the preferred anesthesia type in high-risk patients with dementia. However, as the results were different for different subtypes of dementia, further studies are needed.

  1. Comment:

Abstract, line 19: “This study aimed to investigated incidence…”. Change to either: i) “This study investigated the incidence of dementia”, or ii) This study aimed to investigate the incidence of dementia”.

Answer:

Thank you for your comment. We correct it.

This study aimed to investigate the incidence and risk of dementia according to the surgery type.

  1. Comment:

Abstract, line 24: Capitalise the beginning of the sentence. “… of each surgery type. The incidence rates of dementia…

Answer:

Thank you for your comment. We correct it.

The incidence rates of dementia in the control

  1. Comment

Line 35: Provide a reference for the claim that dementia is the 7th most common cause of death.

Line 37: Provide a reference for the claim that there are 55 million dementia  patients worldwide etc.

Answer:

Thank you for your comment. We correct it.

The reference of two sentence above and next sentence is all the same.

Dementia is currently ranked the seventh cause of death among all diseases, and is one of the main causes of disability and dependence among the elderly worldwide. At present, more than 55 million dementia patients are reported worldwide. Every year, dementia develops in about 10 million patients. Dementia is a deterioration in cognitive function beyond the usual consequences of biological aging [1].

  1. Comment

Line 38: The full stop should occur before the [1] reference.

Answer:

Thank you for your comment. We correct it.

Dementia is a deterioration in cognitive function beyond the usual consequences of bio-logical aging [1].

  1. Comment

Figure 1. and Table 1. CCI has not been defined. Either the Charleson comorbidity index should be spelled out in the figure/table, or an abbreviations caption should be provided below the figure/table.

Answer:

Thank you for comment, we added description.

; CCI, Charlson comorbidity index

The Charlson comorbidity index is weighted index to predict risk of death within 1 year of hospitalization for patients with specific comorbid conditions.

  1. Comment

Line 95: The International Classification of Diseases (ICD) appears to be an established assessment tool. Further, it appears the authors are using the 10th revision of the ICD? Please reference accordingly.

Answer:

Thank you for comment, we added reference.

CDC. International Classification of Diseases,Tenth Revision (ICD-10). Available online: https://www.cdc.gov/nchs/icd/icd10.htm (accessed on March 3, 2022.)

  1. Comment

Line 98: I am unfamiliar with the terms F00, G30, F02, F03. What does the F stand for? What do the values 00, 30, 02, 03 stand for? Could the authors please provide more information about these coding values?

Answer:

Thank you comment. We added description of code in appendix table 1.

F stand for Mental and behavioural disorders

G stand for Diseases of the nervous system

F00: Dementia in Alzheimer’s disease

G30: Alzheimer’s disease

F01: Vascular dementia

F02: Dementia in other diseases classified elsewhere

F03: Unspecified dementia

  1. Comment

Line 102-103: The Charlson comorbidity index appears to be an established assessment tool. Please reference accordingly.

Answer:

Thank you for comment, we added reference.

Comparative Study J Chronic Dis. 1987;40(5):373-83. doi: 10.1016/0021-9681(87)90171-8.

  1. Comment

Line 105 and Appendix Table A1: Can the authors provide more information on the ‘original’ and ‘updated weights’ in Appendix Table A1? Is the original value referring to the ICD-10 and the updated weight referring to the KCD? Further can the authors justify why the weights were updated? For example, what was the reasoning for revising the peripheral vascular disease classifications from a 1 (original weight) to 0 (updated weight)?

Answer:

In the study using the health insurance claim database where comorbidities act as a confounder, comorbidity adjustment holds importance. However, researchers are faced with a myriad of options without sufficient information on how to appropriately adjust comorbidity. To date, no consensus has been formed regarding the appropriate index, look back period, and data range in comorbidity adjustment. Recently, several studies (1-4) suggest that the Charlson comorbidity index be selected when predicting outcomes such as disease events and mortality. So, to adjust comorbidity between two groups, we used the Charlson comorbidity index in this study using KCD diagnostic code which referred to the ICD-10.

(1) Sci Rep. 2020 Aug 13;10(1):13715.

(2) PLoS One. 2015 May 18;10(5):e0127240.

(3) Perit Dial Int. 2017 1-2;37(1):94-102.

(4) Transplant Proc. 2018 May;50(4):1068-1073.

  1. Comment

Line 107 and Appendix Table A2: The authors claim Appendix Table A2 shows the description of ‘time to event’. I am unsure which values refer to ‘time’? Should a time value be provided?

Answer:

I’m very sorry about this confusing expression. Actually, “time to event” means the period from the enrollment date to the event date. Thus, I modified this sentence more clearly in Appendix Table A3.

  1. Comment

Appendix Table A2. It appears this table refers to the i) total number of surgical events, ii) the number of patients who developed a dementia following surgery, categorised according to surgical event, iii) the number of patients who subsequently died, categorised according to surgical event, and iv) the number of patients who were not able to be followed up, categorised according to surgical event? If I have understood this table correctly, then this table could be improved if the categories went across the page (see example attached) and the table provided the number and proportional values (i.e., as a percentage of the total value: e.g., 63% etc). Can the authors clarify the heading of the second column (i.e., “The number of dementia event”) of Table A2? This would imply that all numbers in the second column refer to patients developing dementia?

The authors could consider redesigning the table the following way? (see attached)

Answer:

I totally agreed with your opinion. The appendix table A2 is too confusing; thus, I modified this more clearly and readably in the Appendix Table A3.

  1. Comment

Lines 108-110. The authors state: “The types of surgery were classified into nine categories: musculoskeletal system, skin and soft tissue, digestive surgery, genitourinary surgery, obstetrics and gynecologic surgery, neurosurgery, and circulatory surgery.” There are two missing categories in this sentence: ‘non-anesthesia’ and ‘others’? The authors should define ‘others’ here. The authors should consider bullet pointing the categories. For example, “The types of surgery were classified into nine categories: i) non-anesthesia, ii) musculoskeletal system, iii) etc”.

Answer:

Thank you for comment.

The subjects of this study were patients who underwent surgery through general anesthesia or neuraxial anesthesia. There were no patients without anesthesia. The others were patients who underwent the rest of the surgery except for seven kinds of surgeries above under general or neuraxial anesthesia.

The types of surgery were classified into nine categories: musculoskeletal system, skin and soft tissue surgery, digestive surgery, genitourinary surgery, obstetrics and gynecologic surgery, neurosurgery, circulatory surgery, and others. The others category included patients who underwent surgeries except for the aforementioned seven surgery types under general or neuraxial anesthesia. Moreover, patients who underwent two or more surgeries were assigned to the two or more surgeries category.

  1. Comment

Line 203. There is a double space between ‘were inconsistent’.

Answer:

Thank you for comment. We corrected it

  1. Comment

Line 315. Appendix Table A3. Consider starting the table title on a new page. It is disconnected from the table.

Answer:

Thank you for comment. We corrected it

  1. Comment

Line 337. The References title should be on a new page.

Answer:

Thank you for comment. We corrected it

Round 2

Reviewer 2 Report

The authors have been very willing to improve the manuscript, however, I still find aspects of the study aims confusing. This may be due to differences in English terminology. For example, in the Introduction, the authors state:

'We evaluated the incidence and risk of dementia according to the surgical type in this study, and included Alzheimer's disease and vascular dementia, which are common subtypes of dementia.'

I think the authors are trying to say:

'The primary aims of this study were to i) evaluate if the incidence of developing a dementia was related to anesthesia type (i.e., general vs neuraxial anaesthesia), and ii) determine if dementia rates varied according to surgery type (e.g., musculoskeletal, circulatory etc.). Of the cohort that developed a dementia post surgery, secondary aims of this study were to i) evaluate if the onset of Alzheimer's disease or vascular dementia—the two most common dementia subtypes—was more common relative to a generalised dementia population, and ii) determine the interplay of surgical type on dementia subtype.'

Due to lack of clarity throughout the manuscript, however, I'm not confident that these are, in fact, the research aims of the study—so I will not advise to update accordingly.

I will accept this manuscript in preset form; however, I encourage the authors to try to further clarify the study aims and refer to these aims throughout (i.e., in the results and discussion sections). This will ensure that the study will receive wide readership and citations.

Author Response

Comment

The authors have been very willing to improve the manuscript, however, I still find aspects of the study aims confusing. This may be due to differences in English terminology. For example, in the Introduction, the authors state:

'We evaluated the incidence and risk of dementia according to the surgical type in this study, and included Alzheimer's disease and vascular dementia, which are common subtypes of dementia.'

I think the authors are trying to say:

'The primary aims of this study were to i) evaluate if the incidence of developing a dementia was related to anesthesia type (i.e., general vs neuraxial anaesthesia), and ii) determine if dementia rates varied according to surgery type (e.g., musculoskeletal, circulatory etc.). Of the cohort that developed a dementia post surgery, secondary aims of this study were to i) evaluate if the onset of Alzheimer's disease or vascular dementia—the two most common dementia subtypes—was more common relative to a generalised dementia population, and ii) determine the interplay of surgical type on dementia subtype.'

Due to lack of clarity throughout the manuscript, however, I'm not confident that these are, in fact, the research aims of the study—so I will not advise to update accordingly.

I will accept this manuscript in preset form; however, I encourage the authors to try to further clarify the study aims and refer to these aims throughout (i.e., in the results and discussion sections). This will ensure that the study will receive wide readership and citations.

Answer

Thank you very much for your comment.

We corrected discussion according to your advice.
